# Antifungal Activity of Chemical Constituents from *Piper pesaresanum* C. DC. and Derivatives against Phytopathogen Fungi of Cocoa

**DOI:** 10.3390/molecules26113256

**Published:** 2021-05-28

**Authors:** Luis C. Chitiva-Chitiva, Cristóbal Ladino-Vargas, Luis E. Cuca-Suárez, Juliet A. Prieto-Rodríguez, Oscar J. Patiño-Ladino

**Affiliations:** 1Department of Chemistry, Faculty of Sciences, Universidad Nacional de Colombia, Sede Bogotá, Bogotá 111321, Colombia; lchitiva@unal.edu.co (L.C.C.-C.); lecucas@unal.edu.co (L.E.C.-S.); 2Department of Chemistry, Faculty of Sciences, Pontificia Universidad Javeriana, Sede Bogotá, Bogotá 110231, Colombia; cristoballadino@javeriana.edu.co (C.L.-V.); juliet.prieto@javeriana.edu.co (J.A.P.-R.)

**Keywords:** *Piper pesaresanum*, *Theobroma cacao*, *Moniliophthora roreri*, *Fusarium solani*, *Phytophthora* sp., benzoic acid derivatives, dihydrochalcones, phytosanitary agents

## Abstract

In this study, the antifungal potential of chemical constituents from *Piper pesaresanum* and some synthesized derivatives was determined against three phytopathogenic fungi associated with the cocoa crop. The methodology included the phytochemical study on the aerial part of *P. pesaresanum*, the synthesis of some derivatives and the evaluation of the antifungal activity against the fungi *Moniliophthora roreri*, *Fusarium solani* and *Phytophthora* sp. The chemical study allowed the isolation of three benzoic acid derivatives (**1**–**3**), one dihydrochalcone (**4**) and a mixture of sterols (**5**–**7**). Seven derivatives (**8**–**14**) were synthesized from the main constituents, of which compounds **9**, **10**, **12** and **14** are reported for the first time. Benzoic acid derivatives showed strong antifungal activity against *M. roreri*, of which **11** (3.0 ± 0.8 µM) was the most active compound with an IC_50_ lower compared with positive control Mancozeb^®^ (4.9 ± 0.4 µM). Dihydrochalcones and acid derivatives were active against *F. solani* and *Phytophthora* sp., of which **3** (32.5 ± 3.3 µM) and **4** (26.7 ± 5.3 µM) were the most active compounds, respectively. The preliminary structure–activity relationship allowed us to establish that prenylated chains and the carboxyl group are important in the antifungal activity of benzoic acid derivatives. Likewise, a positive influence of the carbonyl group on the antifungal activity for dihydrochalcones was deduced.

## 1. Introduction

Cocoa is one of the primary products of significant importance in world trade, widely used in food, pharmaceutical and cosmetic industries [1,2]. At the therapeutic level, it stands out for its antioxidant and anti-inflammatory properties and its stimulating action on the central nervous system [3,4]. The production and trade of cocoa beans and their derived products constitute the base of the national economy of several underdeveloped countries [5]. Worldwide, around 6.7 million hectares are estimated, with a cocoa bean production of 4.7 million tons and generating income of approximately 8.6 billion dollars per year. However, production yields are moderate to low in many of the countries where it is cultivated. The reduction in productivity is caused by several factors, among which stand out the lack of technification of the crops, the drastic climatic changes, attack of pests and diseases [6,7].

Plant pathogens often cause various diseases in cocoa plantations, altering the physiological functions of plants and in extreme cases can cause plant death [8]. *Moniliophthora roreri* affects the fruits causing the disease known as moniliasis [9]. *M. perniciosa* affects the growing tissues of the plant (branches, flowers and fruits) causing the disease known as “witch’s broom” [10]. Species of the genus *Phytophthora* cause the disease known as “black ear” [11]. *Ceratocystis fimbriata* causes the disease known as “machete disease”, which affects various vegetative organs and can cause plant death [12]. Additionally, endophytic fungi such as *Fusarium solani* have been reported as phytopathogens when they negatively affect the plant. *Fusarium* species have been reported to cause diseases known as *Fusarium* vascular dieback (FVD) and vascular streak dieback (VSD) in cocoa crops [13,14]. Adequate integrated management of the crop is the most used strategy for the control of these phytopathogenic agents, which combines cultural control methods and the use of commercial pesticides [15]. However, the indiscriminate use of chemical products has led to damages to the environment and health problems for farmers and consumers, as well as the appearance of resistant pathogenic microorganisms [16,17]. Therefore, it is necessary to find effective and safe alternatives for the control of plant pathogens.

In the search for new phytosanitary agents, plants represent an interesting and promising alternative due to the various types of bioactive compounds that they produce, which can be used as control methods or can be the source of inspiration for the development of more active substances [18,19]. *Piper* genus, which belongs to the Piperaceae family, has 1457 species distributed in the tropical and subtropical regions around the world [20]. Extracts, essential oils and chemical constituents from some of their species have presented an antifungal potential for the control of plant pathogens that affect crops of economic importance. Researches have focused mainly on the control of *Aspergillus flavus*, *A. oryzae*, *Botrytis cinerea*, *Cladosporium cladosporioides*, *C. sphaerospermum*, *C. cucumerinum*, *F. oxysporum*, *M. roreri, Lasiodiplodia theobromae, Puccinia recondita, Phytophthora infestans, P. cinnamomi* and *Rhizopus stolonifer* [21,22]. Chemical constituents with antifungal potential have been found in species of *Piper* genus, including amides, flavonoids, alkylbenzenes, terpenoids and benzoic acid derivatives [23]. *P. pesaresanum* C. DC. (Synonym *P. irazuanum* C. DC.) is a species that has been reported in Colombia and Panama, commonly at high altitudes [24]. Hexane and Dichloromethane fractions from the aerial part of this species have been determined to have the ability to inhibit the growth of *Mycosphaerella fijiensis*, an ascomycete fungus that causes black Sigatoka disease in banana crops [25]. In addition, a benzoic acid derivative present in its leaves has been reported with inhibitory activity against acetylcholinesterase [26]. 

The present research describes the antifungal potential of chemical constituents from *P. pesaresanum* and some synthesized derivatives against three phytopathogenic fungi associated with the cocoa crop.

## 2. Results and Discussion

### 2.1. Phytochemical Study

Phytochemical study of the aerial part of *P. pesaresanum* led to the isolation of four compounds, which were identified by spectroscopic methods and by comparison with the data described in the literature. Additionally, a mixture denominated **M-1** was obtained, which was analyzed by GC-MS. The isolated and identified compounds correspond to three derivatives of benzoic acid (4-methoxyinervogenic acid (**1**), nervogenic acid (**2**) and 3-(3′,3′-dimethylallyl-1′-oxo)-5-(3′′,3′′-dimethylallyl)-4-hydroxybenzoic acid (**3**)), a dihydrochalcone (2′,6′-dihydroxy-4′-methoxydihydrochalcone (**4**)), and in the mixture, **M-1** determined the presence of three sterols (campesterol (**5**), stigmasterol (**6**) and γ-sitosterol (**7**)) (Figure 1). This is the first report of compounds **2** to **7** for *P. pesaresanum*, while **1** has been previously reported in its leaves [26]. All compounds have been isolated from other *Piper* species and these results agree with the chemotaxonomy of the genus. In some *Piper* species the presence of benzoic acid derivatives is common, characterized by oxygenated substituents in position 4 and prenylated chains in positions 3 and/or 5 on the aromatic ring [27,28]. The dihydrochalcones reported in *Piper* are characterized by the presence of oxygenated substituents on ring A at positions 2′, 4′, and 6′, while in ring B no substituents are usually observed [29]. The mixture of sterols found in this study has been commonly reported in plants, including *Piper* species [30,31,32].

Compounds **1** to **3** were white crystalline solids, which produced red staining on TLC when sprayed with vanillin/H_2_SO_4_ reagent. The NMR spectra for **1** to **3** showed the typical signals of benzoic acid derivatives reported for species of *Piper* genus [29,33]. NMR analysis for the compound **1** indicated the presence of a 1,3,4,5-tetrasubstituted aromatic ring by the signal in ^1^H-NMR at δ_H_ 7.80 (s, 2H) and signals at δ_C_ 161.2 (C), 133.4 (C), 130.4 (CH) and 125.1 (C) observed in APT experiment. The substituents on the aromatic ring were determined as a carboxyl group located on the position C-1 (signal at δ_C_ 172.2 (C)), a methoxyl group in C-4 (signals at δ_H_ 3.78 (s, 3H) and δ_C_ 61.1 (OCH_3_)) and two groups 3´,3´-dimethylallyl located in the positions C-3 and C-5 (signals at δ_H_ 5.29 (t, *J* = 7.1 Hz, 2H), 3.40 (d, *J* = 7.1 Hz, 4H), 1.76 (s, 6H) and 1.75 (s, 6H) in ^1^H-NMR, together with the signals at δ_C_ 135.3 (C), 122.4 (CH), 28.5 (CH_2_), 25.9 (CH_3_) and 18.0 (CH_3_) in APT). The comparison of the spectroscopic data obtained with those reported in the literature allowed the identification of **1** as 4-methoxynervogenic acid. The comparison of the spectroscopic data with those reported in the literature allowed the identification of **1** as 4-methoxynervogenic acid [33]. Compounds **1** and **2** have a similar NMR profile but in the NMR spectra of **2** it lacks the characteristic signal of the methoxy group. Instead, the presence of a hydroxyl group at position 4 of the aromatic ring was determined. Comparison with the spectroscopic data reported in the literature allowed the identification of **2** as a nervogenic acid [33]. Compound **3** has an NMR profile similar to **2**, but **3** has a carbonyl group in one of the prenylated chains. The characteristic signals in NMR for the oxoprenyl group were assigned as δ_H_ 6.88 (s, 1H), 2.24 (s, 3H), 2.09 (s, 3H) in ^1^H-NMR and signals at δ_C_ 196.0 (C), 159.8 (C), 119.6 (CH), 28.4 (CH_3_) and 21.6 (CH_3_) in the APT experiment. Additionally, the characteristic signal of a chelated proton in ^1^H-NMR spectra was observed (signal δ_H_ 13.79 (s, 1H)). Compound **3** was identified as 3-(3′,3′-dimethylallyl-1′-oxo)-5-(3′′,3′′-dimethylallyl)-4-hydroxybenzoic acid after comparing with the spectroscopic data reported in the literature [33].

Compound **4** was a white amorphous solid, which on TLC produced a yellow fluorescence characteristic for flavonoids after being sprayed with the natural product reagent (NP/PEG) and visualized with 365 nm UV light [34]. NMR analysis for compound **4** indicated the presence of a dihydrochalcone-type flavonoid by the signals at δ_H_ 3.41 (t, *J* = 7.3 Hz, 2H) and 2.98 (t, *J* = 7.3 Hz, 2H) in ^1^H-NMR and signals at δ_C_ 206.2 (C), 46.5 (CH_2_) and 31.3 (CH_2_) in APT. The presence of a methoxyl group at position C-4′ (signals at δ_H_ 3.79 (s, 3H) and δ_C_ 55.8 (OCH_3_)) and two hydroxyl groups at position C-2′ and C-6′ (signal at δ_H_ 11.81 (s, 1H) and δ_C_ 166.9 (C) 165.2 (C)) was established on the aromatic ring. According to the previous analysis and after comparing with the spectroscopic data reported in the literature, **4** was identified as 2′,6′-dihydroxy-4′-methoxydihydrochalcone [35]. The NMR spectra obtained for the isolated compounds are presented in the Appendix A.

The GC-MS analysis of **M-1** allowed the determination of three signals with retention times 28.891, 29.173 and 29.740 min, and integrated area of 20.1, 41.2 and 38.7%, respectively. The comparison of mass spectra and Kovats retention indices (KIs) for each peak with the reports in the literature indicated that the signals correspond to campesterol (**5**), stigmasterol (**6**) and γ-sitosterol (**7**), respectively (Appendix A and Appendix A) [30,31].

### 2.2. Synthesis of Derivates

To establish the influence of the functional groups on the antifungal activity, some derivatives were synthesized from the isolated and identified chemical constituents (Scheme 1). Compound **1** was subjected to methylation, decarboxylation and hydrogenation reactions for the preparation of derivatives **8** to **10**, respectively. Compound **8** was obtained in good yield (80%) by a typical methylation reaction in the presence of methyl iodide [36]. The decarboxylation reaction was carried out in the presence of sodium persulfate, obtaining **9** with a low yield (12%) [37]. Finally, the reduction of the double bonds present in the prenylated chains of **1** was carried out by catalytic hydrogenation in the presence of Pd/C, obtaining compound **10** in good yield (96%) [38]. Compound **2** was subjected to intramolecular oxidative cyclization, in the presence of 2,3-dichloro-4,5-dicyano-1,4-benzoquinone (DDQ), which allowed the formation of chromene **11** (30%) with a yield comparable to those reported for similar reactions [39]. 

Chromanone **12** was synthesized for the first time and with a good yield (89%) through an oxo-Michael intramolecular cyclization from compound **3** [40]. Compound **4** was subjected to methylation and reduction reactions to obtain **13** and **14**, respectively. Methylation was carried out on the phenolic hydroxyls at the 2′ and 6′ positions in ring A in presence of methyl iodide [41]. The reduction of the carbonyl group to methylene was carried out under the typical conditions of a Clemmensen reduction (amalgam of zinc and acid medium) [42]. The NMR spectra obtained for the synthesized compounds are presented in the Appendix A. This is the first report in the literature for compounds **9**, **10**, **12** and **14**, while **8**, **11** and **13** have been previously reported in *Piper* species [33,43,44,45].

### 2.3. Antifungal Activity and Qualitative Structure–Activity Relationship Analysis

Natural and synthetic compounds were evaluated as antifungals against *M. roreri*, *F. solani* and *Phytophthora* sp. by mycelial growth inhibition test. The results of antifungal activity expressed as half-maximal inhibitory concentration (IC_50_) are summarized in Table 1. To the best of our knowledge, this is the first report of the antifungal activity of compounds **1** to **14** against the three phytopathogenic fungi studied. In addition, no reports have been found in the literature of antifungal activity against other phytopathogenic fungi for compounds **1** to **3**, **8** to **10** and **12** to **14**, while antifungal activity has been reported against *C. cladosporoides* and *C. sphaerospermum* for **4** [44], and *Penicillium oxalicum* for **11** [46].

Benzoic acid derivatives **8** (6.7 ± 0.7 µM) and **11** (3.0 ± 0.8 µM) showed high antifungal activity against *M. roreri*, highlighting that the IC_50_ value for **1** is lower than that determined for Mancozeb^®^ (4.9 ± 0.4 µM). Antifungal potential of natural compounds, synthetic derivatives and commercial compounds was used to establish preliminary structure–activity relationships. The importance of the presence of the carboxyl group and prenylated chains could be deduced by comparing the antifungal activity of the benzoic acid derivatives with the commercial compounds evaluated. After comparing the IC_50_ values of compounds **2** and **11** it was possible to establish that the ring closure to obtain chromenes strongly increased antifungal activity, the IC_50_ of **11** being approximately 33 times lower than that of **2**. Moreover, it was found that compound **8** is approximately eight times more active than **1,** indicating that the formation of a methyl ester leads to an increase in antifungal activity. The comparison between **1** and **9** allowed us to determine that, when the carboxyl group is eliminated, the activity is reduced about four times, which indicates that this group is important for the antifungal activity of this type of compound. In the same way, it was established that the phenolic hydroxyl and the carbonyl group are important for the antifungal activity of dihydrochalcone **4**.

Dihydrochalcones and acid derivatives exhibited moderate antifungal activity against *F. solani* with IC_50_ values between 32.5 and 101.8 µM. Preliminary structure–activity analysis suggests that the presence of carboxyl, isoprenyl and hydroxyl groups in the structure of acid derivatives is necessary to exhibit antifungal activity against *F. solani*. Compound **3** was the most active against this pathogen; it is a derivative of acyclic benzoic acid with a free hydroxyl in position 4 and it has isoprenyl and oxoprenyl type chains in positions 5 and 3, respectively. The presence of the methoxyl group in position 4 of the aromatic ring reduces the antifungal activity approximately two times; this effect could be evidenced after comparing the IC_50_ values of compound **1** with those of **2**. Moreover, it was found that the presence of the carbonyl group on the isoprenyl chain enhanced the antifungal activity approximately 1.6 times. In the case of dihydrochalcones, it was established that the presence of the hydroxyl groups on the aromatic ring A and the carbonyl group are determinants for the antifungal activity. Comparing **4** with **3**, it was observed that the methylation of the phenolic hydroxyls of **4** produced a decrease in the antifungal activity of two times. The reduction of the carbonyl group to methylene carried out on compound **4** led to the antifungal activity being reduced three times.

The results of antifungal activity against *Phytophthora* sp. indicate that dihydrochalcones and benzoic acid derivatives show moderate activity, with IC_50_ values between 26.7 and 83.3 µM. Preliminary structure–activity analysis indicated a drastic change in the antifungal activity when is removed the carboxyl group in compound **1** (reducing the IC_50_ by one order of magnitude), which suggests that the presence of this group is very important in the antifungal activity of benzoic acid derivatives. The antifungal activity of the dihydrochalcones is not significantly affected by the reduction of the carbonyl group or the methylation of the phenolic hydroxyls.

## 3. Materials and Methods

### 3.1. General Experimental Procedures

All commercially available reagents employed were used without further purification, while the solvents were technical grade and distilled before use. Thin-layer chromatography (TLC) was performed on SiliaPlate^TM^ alumina plates pre-coated with silica gel 60 F_254_ (SiliCycle^®^ Inc, Quebec, Canada). Vacuum Liquid Chromatography (VLC) was performed on SiliaPlate^TM^ silica gel F_254_ of size 5–20 μm (SiliCycle^®^ Inc, Quebec, Canada). Flash Chromatography (FC) was performed on SiliaFlash^®^ silica gel P_60_ of size 40–63 μm (SiliCycle^®^ Inc, Quebec, Canada). Melting points were recorded on a Thermo Scientific 00590Q Fisher-Johns apparatus (Thermo Scientific^®^, Waltham, MA, USA). NMR measurements were performed on Bruker Advance AC-400 spectrometer (Bruker^®^, Hamburg, Germany) ^1^H-NMR and APT experiments, operating at 400 MHz for ^1^H and 100 MHz for APT; ^1^H-^1^H, direct ^1^H-^13^C and long-range ^1^H-^13^C scalar spin–spin connectivity were established by 2D spectroscopic analysis of the COSY, HMQC and HMBC experiments. Chemical shifts (δ) were reported in part per million (ppm) and coupling constants (J) in Hz. The following abbreviations were used to designate chemical shift multiplicities: s = singlet, d = doublet, t = triplet, q = quartet, m = multiplet, bs = broad singlet. High-resolution mass spectrometry (HRMS) analyses were performed on LC-MS QTOF 9030, Nexera X2 (Shimadzu^®^, Duisburg, Germany) system. The ionization method was ESI operated in positive and negative ion mode. The samples were dissolved in methanol.

GC-MS analysis was performed in electronic impact (EI) mode on a Shimadzu GC 2010 Plus chromatograph with a Zebron ZB-5plus capillary column with a stationary phase of 5% phenyl-95% dimethylpolysiloxane 30 m × 0.25 mm, with ID x 0.25 µm, coupled to a Shimadzu GCMS-TQ 8040 selective mass detector. The injection conditions used were the temperature of the injector 250 °C; the oven temperature was programmed from 60 °C (1 min) at 10 °C/min to 320 °C (5 min) with a split/splitless injection port (split ratio 1:10), carrier gas He: 1 mL/min, constant flow; sample volume 1 μL. The mass spectrometer was operated at 70 eV. Samples were prepared by dissolving 1 mg in 1 mL of hexane. The identification of the chemical constituents was based on the comparison of their mass spectra and experimental calculations of the Kovats retention indices (KIs) compared with those obtained in the NIST 14 database.

### 3.2. Plant Material

The aerial part of the species was collected in Santa Barbara town in the department of Santander (Colombia) at an altitude of approximately 2600 m. The species was determined by the biologist Ricardo Callejas and a specimen of *P. pesaresanum* C. DC. was deposited in the Herbario Nacional Colombiano with voucher number COL-553307.

### 3.3. Extraction and Isolation of Compounds 

The dried and ground aerial part of *P. pesaresanum* (1000 g) was extracted with EtOH 96% by the maceration method at room temperature. The resulting solution was concentrated under a vacuum to obtain 150.5 g of extract. A part of the extract (90.5 g) was fractionated by VLC eluted with solvents of different polarity: Dichloromethane (DCM), methyl acetate (MeOAc), isopropanol (*i-*PrOH) and EtOH:H_2_O (8:2). The DCM fraction (40.4 g) was subjected to FC eluted with a mixture of hexane:EtOAc in increasing polarity (95:5 to 50:50), obtaining 50 fractions that were combined in 10 final fractions according to the study by TLC. Fractions 1 and 2 were combined (17.6 g) and purified by successive FC eluted with hexane:EtOAc (95:5), DCM:EtOAc (85:15) and DCM, obtaining compound **1** (2.2 g). Fractions 3 and 4 (15.6 g) were subjected to purification by successive FC eluted with hexane:EtOAc (97:3), DCM:EtOAc (90:10) and DCM, obtaining compound **2** (60.2 mg). Fractions 5–7 were combined (3.6 g) and subjected to purification by successive FC eluted with hexane:EtOAc (95: 5) and DCM:EtOAc (80:20), obtaining compound **3** (70.5 mg). Fractions 8 and 9 were combined (2.3 g) and subjected to purification by successive FC eluted with hexane:EtOAc (80:20) and DCM:EtOAc (70:30), obtaining compound **4** (1.4 g). Fraction 10 (1.3 g) was purified by successive FC eluted with hexane:EtOAc (98:2), obtaining mixture **M-1** composed by **5**–**7** (80.6 mg), after its characterization by GC-MS.

*4-methoxynervogenic acid* (**1**). White solid, m.p. 80–82 °C. ^1^H-NMR (CDCl_3_, 400 MHz): δ (ppm) 7.80 (s, 2H), 5.29 (t, *J* = 7.1 Hz, 2H), 3.78 (s, 3H), 3.40 (d, *J* = 7.1 Hz, 4H), 1.76 (s, 6H), 1.75 (s, 6H). APT (CDCl_3_, 100 MHz): δ (ppm) 172.2 (C=O), 161.2 (C-4), 135.3 (C-3′, C-3′′), 133.4 (C-3, C-5), 130.4 (C-2, C-6), 125.1 (C-1), 122.4 (C-2′, C-2′′), 61.1 (OMe), 28.5 (C-1′, C-1′′), 25.9 (C-4′, C-4′′), 18.0 (C-5′, C-5′′). The spectroscopic data were consistent with those reported in the literature for 4-methoxynervogenic acid [33].

*Nervogenic acid* (**2**). White solid, m.p. 79–81 °C. ^1^H-NMR (CDCl_3_, 400 MHz): δ (ppm) 7.77 (s, 2H), 5.32 (t, *J* = 7.2 Hz, 2H), 3.38 (d, *J* = 7.1 Hz, 4H), 1.79 (s, 12H). APT (CDCl_3_, 100 MHz): δ (ppm) 172.0 (C=O), 158.0 (C-4), 135.4 (C-3′, C-3′′), 130.6 (C-2, C-6), 127.3 (C-3, C-5), 121.4 (C-2′, C-2′′), 121.2 (C-1), 29.7 (C-1′, C-1′′), 26.0 (C-4′, C-4′′), 18.1 (C-5′, C-5′′). The spectroscopic data were consistent with those reported in the literature for nervogenic acid [33].

*3-(3*′*,3*′*-dimethylallyl-1*′*-oxo)-5-(3*′′*,3*′′*-dimethylallyl)-4-hydroxybenzoic acid* (**3**). Yellow solid, m.p. 178–180 °C. ^1^H-NMR (CDCl_3_, 400 MHz): δ (ppm) 13.79 (s, 1H), 8.48 (s, 1H), 8.03 (s, 1H), 6.88 (s, 1H), 5.34 (t, *J* = 7.3 Hz, 1H), 3.39 (d, *J* = 7.3 Hz, 2H), 2.24 (s, 3H), 2.09 (s, 3H), 1.78 (s, 3H), 1.73 (s, 3H). APT (CDCl_3_, 100 MHz): δ (ppm) 196.0 (C=O), 171.3 (C=O), 166.1 (C-4), 159.8 (C-3′), 136.1 (C-2), 134.0 (C-3′′), 131.6 (C-3), 131.0 (C-6), 120.9 (C-2′′), 119.6 (C-2′), 119.5 (C-5), 118.8 (C-1), 28.4 (C-4′), 27.7 (C-1′′), 25.8 (C-4′′), 21.6 (C-5′), 17.8 (C-5′′). The spectroscopic data were consistent with those reported in the literature for 3-(3′,3′-dimethylallyl-1′-oxo)-5-(3′′,3′′-dimethylallyl)-4-hydroxybenzoic acid [33].

*2′,6′-dihydroxy-4′-methoxydihydrochalcone* (**4**). White solid, m.p. 169–171 °C. ^1^H-NMR (acetone-*d_6_*, 400 MHz): δ (ppm) 11.81 (s, 1H), 7.28 (d, *J* = 7.3 Hz, 4H), 7.20–7.14 (m, 1H), 5.99 (s, 2H), 3.79 (s, 3H), 3.41 (t, *J* = 7.3 Hz, 2H), 2.98 (t, *J* = 7.3 Hz, 2H). APT (acetone-*d_6_*, 100 MHz): δ (ppm) 206.2 (C=O), 166.9 (C-2′), 165.2 (C-6′), 164.0 (C-4′), 142.9 (C-1), 129.3 (C-2, C-6), 129.2 (C-3, C-5), 126.7 (C-4), 105.7 (C-1′), 94.4 (C-3′, C-5′), 55.8 (OMe), 46.5 (C-α), 31.3 (C-β). The spectroscopic data were consistent with those reported in the literature for 2′,6′-dihydroxy-4′-methoxydihydrochalcone [35]. 

### 3.4. Preparation of Derivatives

*Methyl 3,5-bis(3*′*,3*′*-dimethylallyl)-4-methoxybenzoate* (**8**). DMSO (3.0 mL) and KOH 80% (3.0 mL) were added to a round-bottom flask and the resulting mixture was stirred for 30 min at room temperature. Subsequently, compound **1** (100 mg, 0.34 mmol) dissolved in DMSO (3.0 mL) was added, and stirring continued for approximately 15 min. The mixture was cooled in an ice-water bath and then CH_3_I (31.0 μL, 0.51 mmol) was added slowly. The mixture was stirred for 2 h bringing it to room temperature. The reaction was monitored by TLC to verify the disappearance of **1**. Subsequently, ice water was added to the reaction mixture and it was extracted with EtOAc (3 × 15 mL). The combined organic layers were washed with brine (2 × 10 mL), dried over anhydrous Na_2_SO_4_ and evaporated under vacuum. The crude obtained product was purified by FC eluted with hexane:EtOAc (95:5) to give compound **8** (80 mg, 80.0% yield). Compound **8**: Yellow oil, yield 80.0%. ^1^H-NMR (CDCl_3_, 400 MHz): δ (ppm) 7.96 (s, 2H), 5.51 (t, *J* = 7.3 Hz, 2H), 4.09 (s, 3H), 3.97 (s, 3H), 3,61 (d, *J* = 7.2 Hz, 4H), 1.97 (s, 6H), 1.96 (s, 6H). APT (CDCl_3_, 100 MHz): δ (ppm) 167.0 (C=O), 160.3 (C-4), 135.0 (C-3′, C-3′′), 132.9 (C-3, C-5), 129.6 (C-2, C-6), 125.9 (C-1), 122.6 (C-2′, C-2′′), 60.9 (OMe), 51.8 (OMe-ester), 28.4 (C-1′, C-1′′), 25.7 (C-4′, C-4′′), 17.8 (C-5′, C-5′′). The spectroscopic data were consistent with those reported in the literature for methyl 3,5-bis(3′,3′-dimethylallyl)-4-methoxybenzoate [33].

*2,6-bis(3*′*,3*′*-dimethylallyl)-1-methoxybenzene* (**9**). Sodium persulfate (81.0 mg, 0.34 mmol) was added to a solution of compound **1** (100 mg, 0.34 mmol) in EtOH (5.0 mL). The reaction mixture was stirred for approximately 5 days at 60 °C, being continuously monitored by TLC until the disappearance of **1**. After this period, it was extracted with DCM (4 × 25 mL) and the combined organic layer was dried over anhydrous Na_2_SO_4_. The crude product resulting from the vacuum concentration was purified by FC eluting with DCM: AcOEt (9: 1), to give **9** (10 mg, 12.0% yield). Compound **9**: White solid, m.p. 130–132 °C. ^1^H-NMR (CDCl_3_, 400 MHz): δ (ppm) 7.19–7.01 (m, 3H), 5.29 (t, *J* = 7.1 Hz, 2H), 3.76 (s, 3H), 3.38 (d, *J* = 7.1 Hz, 4H), 1.75 (s, 12H). APT (CDCl_3_, 100 MHz): δ (ppm) 160.4 (C-1), 135.1 (C-3′, C-3′’), 133.1 (C-2, C-6), 129.7 (C-3, C-5), 125.8 (C-4), 122.7 (C-2′, C-2′′), 61.1 (OMe), 28.6 (C-1′, C-1′′), 25.9 (C-4′, C-4′′), 18.1 (C-5′, C-5′′). HRMS (ESI) calc. for C_17_H_25_O [M + H]^+^: 245.1899, found: 245.1896. 

*3,5-diisopentyl-4-methoxybenzoic acid* (**10**). Pd/C (4.60 mg, 0.04 mmol) was added to a solution of compound **1** (100 mg, 0.34 mmol) in anhydrous MeOH (4.0 mL) and its atmosphere was saturated with molecular hydrogen. The reaction mixture was stirred at room temperature for 40 h and continuously monitored by TLC until the disappearance of **1**. The resulting mixture was filtered on quantitative Whatman paper and the residue was washed with DCM (2 × 20 mL). The combined organic layer was dried over anhydrous Na_2_SO_4_ and concentrated under a vacuum. The crude product was purified by FC eluting with hexane: EtOAc (97: 3) to give **10** (97 mg, 96.0% yield). Compound **10**: Yellow oil, ^1^H-NMR (CDCl_3_, 400 MHz): δ (ppm) 7.82 (s, 2H), 3.79 (s, 3H), 2.75–2.60 (m, 4H), 1.65 (dt, *J* = 6.9, 6.5 Hz, 2H), 1.53 (q, *J* = 6.9, 6.5 Hz, 4H), 0.98 (s, 6H), 0.96 (s, 6H). APT (CDCl_3_, 100 MHz): δ (ppm) 172.6 (C=O), 161.4 (C-4), 136.6 (C-3, C-5), 130.1 (C-2, C-6), 124.9 (C-1), 61.4 (OMe), 40.0 (C-2′, C-2′′), 28.3 (C-3′, C-3′′), 27.9 (C-1′, C-1′′), 22.7 (C-4′, C-4′′), 22.6 (C-5′, C-5′′). HRMS (ESI) calc. for C_18_H_27_O_3_ [M-H]^−^: 291.1965, found: 291.1952.

*2,2-dimethyl-8-(3*′*,3*′*-dimethylallyl)-2H-1-chromene-6-carboxylic acid* (**11**). DDQ (49.94 mg, 0.22 mmol) was added to a solution of compound **2** (32 mg, 0.11 mmol) in toluene (8.0 mL). The reaction was stirred for 10 h at room temperature and continuously monitored by TLC until the disappearance of **2**. The crude product resulting from the vacuum concentration directly was purified by FC eluted with DCM:EtOAc (7:3) to give **11** (10 mg, 30.0% yield). Compound **11:** Yellow solid, m.p. 170–172 °C. ^1^H-NMR (CDCl_3_, 400 MHz): δ (ppm) 7.75 (s, 1H), 7.61 (s, 1H), 6.35 (d, *J* = 9.9 Hz, 1H), 5.65 (d, *J* = 11.4 Hz, 1H), 5.28 (t, *J* = 7.5 Hz, 1H), 3.29 (d, *J* = 6.9 Hz, 2H), 1.74 (s, 6H), 1.46 (s, 6H). APT (CDCl_3_, 100 MHz): δ (ppm) 171.0 (C=O), 155.0 (C-8a), 132.8 (C-3′), 131.9 (C-7), 130.9 (C-3), 129.5 (C-8), 126.8 (C-5), 122.2 (C-4), 122.1 (C-2′), 121.0 (C-6), 120.6 (C-4a), 77.6 (C-2), 28.5 (C-9,10), 28.3 (C-1′), 25.9 (C-4′), 18.0 (C-5′). The spectroscopic data were consistent with those reported in the literature for 2,2-dimethyl-8-(3′,3′-dimethylallyl)-2H-1-chromene-6-carboxylic acid [46].

*2,2-dimethyl-8-(3’,3’-dimethylallyl)-4-oxochroman-6-carboxylic acid* (**12**). Compound **3** (40.2 mg, 0.13 mmol) and a solution of NaOH 10% (2.0 mL, 0.01 mmol) was added into a round-bottomed flask. The resulting mixture was stirred for 12 h at room temperature and after this period it was neutralized by adding drops of HCl 10% and extracted with AcOEt (3 × 20 mL). The combined organic layer was dried over anhydrous Na_2_SO_4_ and concentrated under vacuum. The crude product obtained was purified by FC eluted with DCM:EtOAc (8:2) to give **12** (35 mg, 89.0% yield). Compound **12**: Yellow solid, m.p. 120–123 °C. ^1^H-NMR (CDCl_3_, 400 MHz): δ (ppm) 8.51 (s, 1H), 8.03 (s, 1H), 5.26 (t, *J* = 6.7 Hz, 1H), 3.34 (d, *J* = 7.0 Hz, 2H) 2.77 (s, 2H), 1.76 (s, 3H) 1.74 (s, 3H), 1.49 (s, 6H). APT (CDCl_3_, 100 MHz): δ (ppm) 191.9 (C=O), 170.3 (C=O), 157.8 (C-8a), 133.8 (C-3′), 131.2 (C-7), 130.2 (C-8), 127.8 (C-5), 121.4 (C-6), 121.2 (C-2′), 119.8 (C-4a), 80.3 (C-2), 48.6 (C-3), 28.4 (C-1′), 26.9 (C-9, C-10), 25.9 (C-4′), 18.1 (C-5′). HRMS (ESI) calc. for C_17_H_19_O_4_ [M-H]^−^: 287.1288, found: 287.1271. 

*2*′*,4*′*,6*′*-trimethoxydihydrochalcone* (**13**). Anhydrous K_2_CO_3_ (400 mg, 1.11 mmol) and CH_3_I (2.28 mL, 3.51 mmol) were added to a solution of compound **4** (100 mg, 0.36 mmol) in acetone (10 mL). The reaction was refluxed with continuous stirring for a period of 5 days and continuously monitored by TLC until the disappearance of **4**. After the reaction, the excess CH_3_I was removed under vacuum. The residue was washed with water and brine. Then it was extracted with EtOAc (3 × 20 mL), dried over anhydrous Na_2_SO_4,_ and concentrated under vacuum. The obtained product was purified by FC eluted with DCM:EtOAc (9:1) to give **13** (80 mg, 80.0% yield). Compound **13**: white solid, m.p. 159–160 °C. ^1^H-NMR (CDCl_3_, 400 MHz): δ (ppm) 7.21 (dd, *J* = 15.0, 8.0 Hz, 5H), 6.08 (s, 2H), 3.81 (s, 3H), 3.74 (s, 6H), 3.48 (t, *J* = 7.1 Hz, 2H), 3.02 (t, *J* = 7.1 Hz, 2H). APT (CDCl_3_, 100 MHz): δ (ppm) 203.3 (C=O), 162.0 (C-4′), 157.9 (C-2′, C-6′), 141.3 (C-1), 128.1 (C-5, C-3), 128.0 (C-2, C-6), 125.5 (C-4), 112.3 (C-1′), 90.2 (C-3′, C-5′), 55.5 (OMe-2′,6′), 55.2 (OMe-4′), 46.0 (C-α), 29.7 (C-β). The spectroscopic data were consistent with those reported in the literature for 2′,4′,6′-trimethoxydihydrochalcone [47].

*1-(2*′*,6*′*-dihydroxy-4*′*-methoxyphenyl)-3-phenylpropane* (**14**). Initially, the zinc amalgam was prepared by adding in a reaction flask zinc powder (1000 mg, 15.10 mmol), HgCl_2_ (100 mg, 0.36 mmol), H_2_O (1.5 mL) and HCl 37% (50.0 μL). The mixture was stirred for 5 min and filtered to obtain the amalgam. Subsequently, on the zinc amalgam was added H_2_O (10.0 mL), HCl 37% (5.0 mL) and compound **4** (250 mg, 0.91 mmol). The reaction was refluxed and stirred for a period of 10 h, adding dropwise 0.5 mL of HCl 37% every 2 h during the reaction time. The reaction was cooled on an ice-water bath and filtered to remove excess amalgam. The solution was extracted with DCM (3 × 20 mL). The combined organic layers were dried with Na_2_SO_4_ and concentrated under vacuum. The crude product was purified by FC eluted with DCM:EtOAc (95:5) to give **14** (17 mg, 10.0% yield). Compound **14**: White solid, m.p. 80–82 °C. ^1^H-NMR (acetone-*d_6_*, 400 MHz): δ (ppm) 8.02 (s, 2H), 7.39–7.15 (m, 5H), 5.99 (s, 2H), 3.79 (s, 3H), 3.41 (t, *J* = 7.1 Hz, 2H), 2.98 (t, *J* = 7.1 Hz, 2H), 2.20 (q, *J* = 7.1 Hz, 2H); APT (acetone-*d_6_*, 100 MHz): δ (ppm) 161.4 (C-4′), 158.2 (C-2′, C-6′), 142.3 (C-4), 128.7 (C-6, C-8), 128.4 (C-5, C-9), 126.7 (C-7), 111.0 (C-1′), 96.4 (C-3′, C-5′), 55.5 (OMe), 35.8 (C-3), 31.3 (C-2), 24.6 (C-1). HRMS (ESI) calc. for C_16_H_17_O_3_ [M + H]^+^: 259.1328, found: 259.1321. 

### 3.5. Fungal Strains

Strains of *M. roreri*, *F. solani* and *Phytophthora* sp. used in this study were taken from the fungal strain collection of the Bioassays Laboratory of the Pontificia Universidad Javeriana. Microorganisms were kept under controlled temperature conditions at 24 ± 1 °C under dark conditions.

### 3.6. Mycelial Growth Inhibition Assay

To evaluate the antifungal activity of the isolated and synthetic compounds, a mycelial growth inhibition assay was used [48]. Additionally, three commercial compounds (phenol (**15**), benzoic acid (**16**) and *p*-hydroxybenzoic acid (**17**)) were evaluated to contribute to the preliminary structure–activity study. Stock solutions of each compound were prepared in ethanol with concentrations of 1000, 500, 250, 125, 62.5, 31.3, 15.6 and 7.8 μg/mL. Afterward, in a test tube were mixed 460 μL of each stock solution and 1840 μL of Potato Dextrose Agar (PDA) medium. Moreover, 500 μL of the remained mixture was applied in each well of a 12 wells box until solidified. The final effective concentrations in each well were: 200, 100, 50, 25, 12.5, 6.3, 3.1 and 1.6 μg/mL. Inoculation with the fungus was performed, locating in the center of each well a 2 mm plug from a fungus strain grown in PDA for 20 days for *M. roreri*, 4 days for *F. solani* and 10 days for *Phytophthora* sp. Mancozeb^®^ was used as the positive control for all fungi and a solution of EtOH 2% was used as the negative control. The mycelial growth of each microorganism in treatments and controls was determined as measures of the area through the free Image-J software (Java Version 1.8.0, National Institutes of Health (NIH), Bethesda, USA). For each concentration evaluated, four (4) replicates and two independent assays were performed. The calculation of the percentage of inhibition was performed using the formula %I = (C-T/T) × 100, where C = growth area in the control and T = growth area in the treatment [49]. IC_50_ values were estimated by nonlinear regression analysis in software IBM SPSS statistics.

### 3.7. Statistical Analysis 

The statistical analysis of the data was made using the software IBM SPSS Statistics 27 (IBM Version 27.0 San Diego, CA, USA). One-way analysis of variance (ANOVA) was used for multiple group comparisons. The Student’s *t*-test was used for a single comparison between two groups. All data reported are the mean two replicates ± SD, the statistical significance considered was *p* ≤ 0.05.

## 4. Conclusions

This study contributes to the chemistry of *P. pesaresanum* through the isolation and identification of three benzoic acid derivatives (**1–3**), a dihydrochalcone (**4**) and a mixture of sterols (**5–7**), of which **2** to **4** are reported for the first time in the species. Additionally, seven derivatives (**8–14**) were synthesized from natural compounds, four of them being reported for the first time (**9**, **10**, **12** and **14**). Finally, the preliminary structure–activity study derived from benzoic acid and dihydrochalcones contributes to research focused on the control of phytopathogenic fungi. The results of this research indicate the potential of benzoic acid derivatives as possible biocontrol agents for the main diseases affecting cocoa crops. 

## Data Availability

Data sharing not applicable.

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
