# Peer review of "Antifungal Activity of Chemical Constituents from Piper pesaresanum C. DC. and Derivatives against Phytopathogen Fungi of Cocoa"

_molecules, 2021, doi:10.3390/molecules26113256_

Round 1

Reviewer 1 Report

The manuscript Antifungal activity of chemical constituents from Piper pesaresanum C. DC. and derivatives against phytopathogen fungi of cocoa is interesting but certain points required more attention by the authors 

among these points:

1- The abstract should be modified to give more details about the methods used 

2- The authors should submit a diagrammatic scheme for isolation in the supplementary 

3- The yields for the synthesis scheme should be given on the figure

4- Although the authors mentioned that they carried out a structure activity relationship study but nothing could be traced regarding the effect of the functional groups change in the discussion 

5- Statistical analysis is very poor especially when the authors compared between the isolated and synthesized compounds 

6- Identification of the sterol fraction with simple matching with the database and mass fragmentation pattern is very limited either authentic should be used or KIs should be calculated and compared with the reported values.

7- The manuscript should be checked by an English native speaker to improve the language used 

Author Response

Dear Editor and Reviewer 1,

We want to thank you for reading and reviewing our manuscript. Your comments are very valuable and were helpful to us. Below, we indicate the changes and adjustments made.

Note: In the manuscript all the changes made are indicated in red. In each of the responses to the comments, the lines to review are indicated.

1- The abstract should be modified to give more details about the methods used.

The abstract was adjusted to clarify the methodology carried out according with the observations.

2- The authors should submit a diagrammatic scheme for isolation in the supplementary.

The requested diagram was added in Figure S1 of the supplementary information.

3- The yields for the synthesis scheme should be given on the figure.

The yields were included in the synthesis scheme according to the reviewer suggestion.

4- Although the authors mentioned that they carried out a structure activity relationship study but nothing could be traced regarding the effect of the functional groups change in the discussion.

In section 2.3 you can find the discussion of the structure-activity relationship, as well as the effect of the modification of the main functional groups on antifungal activity. This discussion is found in lines 190 to 230 of the manuscript.

5- Statistical analysis is very poor especially when the authors compared between the isolated and synthesized compounds.

The statistical analysis was complemented by a one-way analysis of variance (ANOVA) to compare the different multiple groups, and a paired Student's t test was used for a single comparison between groups. Statistical significance was considered at P < 0.05. In addition, the correction of the values of the standard deviation (SD) for the concentration in µM was carried out.

6- Identification of the sterol fraction with simple matching with the database and mass fragmentation pattern is very limited either authentic should be used or KIs should be calculated and compared with the reported values.

The tentative identification of the sterol mixture was carried out by comparing their mass spectra and the Kovats retention indices (KIs), with the NIST 14 databases and reports in the scientific literature. To calculate the KIs, the retention times of n-paraffins (C10-C35) analyzed under the same chromatographic conditions as the samples were used. The calculation was made using equation 1. See Table S1 and Figure S31 of the supplementary material.

7- The manuscript should be checked by an English native speaker to improve the language used.

English grammar was reviewed and corrected throughout the document by a professional in language editing.

We hope that we have made the adjustments properly and that the manuscript has improved.

We greatly appreciate your attention, and we remain attentive to your comments.

Kind regards,

MSc. Luis Carlos Chitiva Chitiva

Universidad Nacional de Colombia

E-Mail: [email protected]

MSc. Cristóbal Ladino Vargas

Pontificia Universidad Javeriana

E-Mail: [email protected]

PhD. Luis Enrique Cuca Suárez

Associate Professor, Department of Chemistry

Universidad Nacional de Colombia

E-Mail: [email protected]

PhD. Juliet Angélica Prieto Rodríguez

Assistant Professor, Department of Chemistry

Pontificia Universidad Javeriana

E-Mail: [email protected]

PhD. Oscar Javier Patiño Ladino

Assistant Professor, Department of Chemistry

Universidad Nacional de Colombia

Corresponding author E-Mail: [email protected]

Reviewer 2 Report

The manuscript entitled „Antifungal activity of chemical constituents from Piper pesaresanum C. DC. and derivatives against phytopathogen fungi of cocoa“ by Chitiva-Chitiva et al., describes the antifungal potential of chemical constituents from P. pesaresanum and 12 some synthesized derivatives. The authors have done a fairly large amount of work and received a significant number of results. The idea of research was very good. Conclusions adequate to the conducted research. Several changes are recommended, and some clarifications are required.

Line 26 plants/fungi names must be italicized.

Line 46, line76 – extra point.

Line 68 Rhyzopus stolonifera correct Rhizopus stolonifer.

Line 175 Can you explain „mean inhibitory concentration“? What do you mean?

Line 181 P. pesaresanum italicized.

In Table 1. What do you mean “CI50”? In whole text IC50? Is it possible to express IC50 in two units simultaneously - µg/mL and µM? Furthermore, the units of the IC50 values should be double-checked.

Line 404 M roreri, F solani – the point after genus name is required.

In section 3.6 The temperature was too high to keep microorganisms.

There is no information in section 3.7 on how you calculated IC50 values. Only the calculation of the percentage of inhibition is done. Can you explain?

Author Response

Dear Editor and Reviewer 2,

We want to thank you for reading and reviewing our manuscript. Your comments are very valuable and were helpful to us. Below, we indicate the changes and adjustments made.

Note: In the manuscript all the changes made are indicated in red. In each of the responses to the comments, the lines to review are indicated.

Line 26 plants/fungi names must be italicized.

We perform the correction of the plants/fungi names to italicized.

Line 46, line76 – extra point.

We perform the spelling correction.

Line 68 Rhyzopus stolonifera correct Rhizopus stolonifer.

We perform the spelling correction.

Line 175 Can you explain „mean inhibitory concentration“? What do you mean?

The mean inhibitory concentration is a measure of potency that indicates the amount of a substance to inhibit a biological process by half or 50%. The term mean inhibitory concentration was changed to the term half-maximal inhibitory concentration (IC50).

Line 181 P. pesaresanum italicized.

We perform the correction of the plant name to italicized.

In Table 1. What do you mean “CI50”? In whole text IC50? Is it possible to express IC50 in two units simultaneously - µg/mL and µM? Furthermore, the units of the IC50 values should be double-checked.

We perform the spelling correction from CI50 to IC50 According to several reports in the literature if it is possible to express the IC50 in two units at the same time. The most advisable thing is to express it in µM, but in ours we are reporting the two units because the tests were mounted in µg/mL and then, considering the molecular weight, it was calculated in µM.

The concentrations were verified twice where two independent tests were set up. It was forgotten to indicate it, but the correction was made in the methodological part lines 430 and 431.

Line 404 M roreri, F solani – the point after genus name is required.

We perform the spelling correction.

In section 3.6 The temperature was too high to keep microorganisms.

According to literature reports, the temperature for the maintenance of the microorganisms evaluated is within the ideal range. For M. roreri, temperatures between 18 to 28 °C have been reported, for F. solani temperatures of 25 °C have been reported and for Phytophthora sp. temperatures between 20 to 25 °C have been reported. The reports can be found in the following articles: 

Sánchez, F., & Garcés, F. (2012). Moniliophthora roreri (Cif y Par) Evans et al. in the crop of cocoa. Scientia Agropecuaria, 3, 249-258.

Migliorini, P., Milanesi, P. M., Mezzomo, R., Maciel, C. G., & Brião Muniz, M. F. (2018). Fusarium solani and F. oxysporum: Etiological agents of damping off in crambe. Brazilian Journal of Agricultural Sciences, 13(1).

Shelley, B. A., Luster, D. G., Garrett, W. M., McMahon, M. B., & Widmer, T. L. (2018). Effects of temperature on germination of sporangia, infection and protein secretion by Phytophthora kernoviae. Plant pathology, 67(3), 719-728.

There is no information in section 3.7 on how you calculated IC50 values. Only the calculation of the percentage of inhibition is done. Can you explain?

In section 3.7 the information on how the IC50 was calculated was included. For the calculation of the IC50, the values of the percentage of inhibition were used which were entered into the SPSS statistics program and by means of a non-linear regression analysis, the percentage of inhibition vs the logarithm of the concentration was plotted to finally calculate the IC50. This information was registered in the manuscript between lines 434 to 435.

We hope that we have made the adjustments properly and that the manuscript has improved.

We greatly appreciate your attention, and we remain attentive to your comments.

Kind regards,

MSc. Luis Carlos Chitiva Chitiva

Universidad Nacional de Colombia

E-Mail: [email protected]

MSc. Cristóbal Ladino Vargas

Pontificia Universidad Javeriana

E-Mail: [email protected]

PhD. Luis Enrique Cuca Suárez

Associate Professor, Department of Chemistry

Universidad Nacional de Colombia

E-Mail: [email protected]

PhD. Juliet Angélica Prieto Rodríguez

Assistant Professor, Department of Chemistry

Pontificia Universidad Javeriana

E-Mail: [email protected]

PhD. Oscar Javier Patiño Ladino

Assistant Professor, Department of Chemistry

Universidad Nacional de Colombia

Corresponding author E-Mail: [email protected]

Round 2

Reviewer 1 Report

The authors responded positively with most of the raised points

Well done